# Factors Correlated with Smoking Cessation Success in Older Adults: A Retrospective Cohort Study in Taiwan

**DOI:** 10.3390/ijerph16183462

**Published:** 2019-09-18

**Authors:** Chih-Po Chang, Wei-Hsin Huang, Ching-Hui You, Lee-Ching Hwang, I-Jung Lu, Hsin-Lung Chan

**Affiliations:** 1Department of Family Medicine, Mackay Memorial Hospital, Taipei 106, Taiwan; chihpo78242002@gmail.com (C.-P.C.); whh.5881@gmail.com (W.-H.H.);; 2Department of Family Medicine, Taipei Municipal Wanfang Hospital, Taipei Medical University, Taipei 116, Taiwan; jeanyou1012@gmail.com; 3Mackay Medical College, New Taipei City 252, Taiwan; 4Department of Leisure and Recreation Management, Chihlee University of Technology, New Taipei City 252, Taiwan; ijunglu@gmail.com

**Keywords:** nicotine replacement therapy, smoke cessation, predictor, varenicline, older smoker

## Abstract

Smoking cessation in the elderly is very important. This study aims to explore the success rate of smoking cessation in the elderly and the factors that predict the success of smoking cessation. We collected data from smokers ≥60 years who visited a medical center in Taiwan during 2017. All patients were prescribed either varenicline or nicotine replacement therapy (NRT) for smoking cessation. The participants were asked about their smoking status after treatment. In total, 129 participants were enrolled. The three- or six-month point abstinence rate was 48.1%. No significant difference was found among baseline characteristics (including age, gender, underlying diseases, smoking duration, daily consumption amount of cigarette, carbon monoxide concentration, Fagerström test for nicotine dependence scores, and treatment method) between quitters and non-quitters, except for the type of medication used. The proportion of quitters using varenicline was significantly higher than that of non-quitters. Multivariate regression analyses showed that the patients who received varenicline were 3.22 times more likely to quit smoking than those who received NRT. Therefore, we suggest that varenicline use may help in smoking cessation in older adults, compared to NRT. Other baseline characteristics may not affect the success rate of smoking cessation in this population.

## 1. Introduction

Tobacco use has a significant negative public health impact by reducing life expectancy by about 15 years [1,2], with 70% of all smoking-related deaths occurring in people older than 60 years [3]. A previous study reveals that older smokers are two times more likely to die from cardiovascular disease than non-smokers, their risk of acute coronary events is double, and their risk of stroke is 1.5-fold higher that of non-smokers [4]. According to statistics from the National Health Service of Taiwan in 2017, about 20% of smokers were older than 60 years, making smoke cessation in the elderly an important issue. 

Smoking cessation has proven benefits for the elderly, including increased life expectancy by about three years at the age of 60 [5], and reduced risk of coronary heart disease and stroke between the ages of 65 and 79 [6]. Therefore, age should not be an obstacle to smoking cessation. On the contrary, physicians should encourage older smokers to quit smoking and play an important role in the smoking cessation process.

Plenty of studies have focused on the success factors of smoking cessation in the general population, while few studies have discussed that in the elderly. The success factors reported include more chronic illness, smoking for shorter duration, higher daily cigarette amounts, being a nondrinker, being married to a non-smoking spouse, or living without other household smokers [7,8,9,10,11]. However, as the Evaluating Adverse Events in a Global Smoking Cessation Study (EAGLES) suggests [12], the role of medications is becoming increasingly important for smoking cessation, which is not considered as a confounder in previous studies. Therefore, our study used drug type as a variable to analyze smoking cessation factors in older adults. 

The purpose of our study was to identify factors associated with smoking cessation success among the elderly to provide effective information in an outpatient care unit.

## 2. Materials and Methods

### 2.1. Study Population/Subjects

We performed a hospital-based, retrospective, cohort study from records of patients visiting the Mackay Memory Hospital in Taiwan from 1 January to 31 December 2017. We collected data from records of patients older than 60 years with national health insurance and who smoked at least 10 cigarettes per day or who had the Fagerström test for cigarette dependence (FTND) scores ≥ 4 [13]. All participants were outpatients who visited family physicians or were referred to our outpatient department by other clinics. We reviewed the patients’ clinical charts, collecting background information, looking for chronic diseases, including hypertension, diabetes mellitus, hyperlipidemia, heart disease, lung disease, neurological disease, cancer, or a psychological disease. All patients were prescribed either varenicline or nicotine replacement therapy (NRT) for smoking cessation.

### 2.2. Data Collection and Outcome Measures

The basic demographic data on the clinical charts including age, gender, body weight, medical history, and smoking history were obtained at the first outpatient clinic visits. Participants were asked about their smoking habits, including daily consumption of cigarette, smoking year, and smoking pack-year. The Fagerström test of nicotine dependence (FTND) was used to assess their smoking dependence (scores range from 0 to 10, with higher scores representing greater nicotine dependence). In general, smokers with FTND ≥ 7 may have strong withdrawal symptoms and relapse early [14]. Exhaled carbon monoxide (CO) levels were assessed with a carbon monoxide monitor.

Treatment methods were classified as pharmacotherapy with or without individual counseling by professional counselors. At the first visit of clinic, patients could choose to either visit physicians for a short advice session and pharmacotherapy or visit a professional counselor for a 30-minute tutorial on smoking cessation followed by visiting our physician for a short advice session and pharmacotherapy. The drug types included varenicline and NRT. NRT contained nicotine patches and gum. As for the choice of drug type, physicians first excluded those with contraindications (i.e., patients with previous allergic reaction), and if there were any concerns, avoided prescriptions for specific drugs. For example, patients with mental illness who have suicidal ideation or with a history of seizures do not take varenicline, patients with dental problems or temporomandibular joint disease are not eligible for nicotine gum and those with skin allergies or dermatologic conditions (e.g., psoriasis, eczema, atopic dermatitis) do not use nicotine patches. Then, the differences in drug types were explained and discussed with the patients to select the most appropriate drug based on their preferences. One treatment course lasted a maximum of eight weeks. Participants who were still smoking after the first eight weeks received another treatment course for smoking cessation. Participants who restarted smoking were also advised to undergo another treatment course. A maximum of two courses per year were given.

The participants were interviewed by telephone to collect information about their smoking status three months and six months after the treatment. Those who self-reported as having stopped smoking at the three- or six-month calls were assumed to have successfully stopped smoking.

The primary study outcome was the factors that affect the success of smoking cessation, whereas the secondary study outcome was the smoking cessation rate of the elderly patients.

### 2.3. Statistical Analysis

We used multivariate logistic regression to investigate the association between smoking cessation and different factors. Data are expressed as means ± SDs for continuous variables, and as percentages for categorical variables.

We performed Student *t*-tests to determine whether the differences in continuous variables were significant between quitters and non-quitters, and between participants receiving varenicline and NRT. In addition, we used chi-square tests to detect significant differences among the categorical variables between quitters and non-quitters.

We adjusted the statistical models as follows: Model 1 was a univariate logistic regression model to estimate characteristics by odds ratios (ORs) and 95% confidence intervals (CIs) for three-month or six-month point-prevalence abstinence in quitters and non-quitters. Model 2 was a multivariate logistic regression adjusted for age, gender, treatment method, hypertension, daily cigarette amount, smoking year, presence of an FTND score ≥ 7, exhaled carbon monoxide (CO) value ≥ 18 ppm (third quartile of CO value), and the drug type (varenicline or NRT).

We conducted all statistical analyses using the SAS software version 9.4 (SAS Institute, Cary, NC). Our criterion for statistical significance was *p-*value < 0.05. Our statistical tests were two-tailed, and we considered *p*-value < 0.05 statistically significant.

### 2.4. Ethical Approval

The Institutional Review Board approved the study protocols at the Mackay Memory Hospital in Taipei city, Taiwan (application number, 17MMHISO049).

## 3. Results

Table 1 presents the demographic and smoking-related characteristics of the patients. For this cohort study, we analyzed data from 129 patients who met the inclusion criteria (109 men (84.5%) and 20 (15.5%) women). The patients’ mean age was 65.9 ± 5.7 years. They smoked on average 21.8 ± 11.1 cigarettes per day. The average smoking years was 40.9 ± 11.1 years, and 72.9% of patients smoked more than 30 pack-years. The mean nicotine dependence score was 6.2 ± 2.6, but 45.7% of patients had scores higher than 7. The average exhaled CO concentration was 14.2 ± 8.5 (ppm), and the third quartile of CO level in this group was 18 ppm. The highest proportion of comorbidity was hypertension (50.4%). In all, 101 (78.3%) patients received varenicline and 28 (21.7%) received NRT. The average treatment duration was 6.0 ± 4.7 weeks. Only 56 (43.4%) patients received pharmacotherapy alone, while 73 (56.6%) patients received both pharmacotherapy and individual counseling. In addition, we found no significant differences in terms of gender, body weight, daily consumption amount of cigarette, smoking pack-years, exhaled CO concentration, FTND score, treatment method, and the presence of underlying systemic disease between the patients who received varenicline and those who received NRT.

Overall, 62 patients quit smoking, and 67 patients failed to quit. The smoking cessation rate was 48.1%. We found no significant difference among the baseline characteristics between quitters and non-quitters except for the type of medication. There was no difference between the two groups in terms of age, gender, body weight, smoking duration, daily consumption amount of cigarette, FTND score, exhaled CO concentration, treatment method, and presence of underlying systemic disease, including diabetes mellitus, hypertension, or heart disease (Table 2). We found the proportion of patients using varenicline in the quitter group was significantly higher than the proportion in the non-quitter group (*p* = 0.02).

After adjusting for age, gender, treatment method, presence of hypertension, daily consumption amount of cigarette, smoking year, FTND score ≥ 7, and CO concentration ≥ 18 ppm, the patients who received varenicline were 3.22 times more likely to quit smoking than those who received NRT (OR, 3.22; 95%CI, 1.23–8.43) (Table 3).

## 4. Discussion

Our results showing the smoking cessation rate in the elderly in our hospital was 48.1%, which was close to the success rate of another study [15], but higher than the smoking cessation rate among individuals of all ages in Taiwan (28.5%) during 2017 [16]. One explanation for this high abstinence rate in our study is that smokers >60 years of age are more likely to have stronger motivation to quit than young smokers by three reasons. First, our patients came to our clinic pursuing smoking cessation treatment, either due to having more chronic diseases, or referral from other clinical physicians due to deterioration of chronic disease [17,18]. Second, the high motivation for older populations is to set a good example for their children or grandchildren [19]. Third, smoking cessation is difficult when smokers are constantly exposed to the environment of smoking by their colleagues, and older smokers have less environmental temptations after retirement.

According to Table 1, the patients’ baseline conditions, including the severity of nicotine dependence, did not affect the choice of medications prescribed by physicians. However, we found that varenicline was more prescribed than NRT (78.3% vs. 21.7%) in clinical settings. The cause of prescribing NRT less frequently may be due to the skin adverse events reported in elderly or the avoidance of chewing gum use in patients with dental problem. Besides, the elderly group were more acceptable to oral medication since they had more underlying comorbidities treated with oral medications, which decreased the barrier of varenicline use on smoking cessation. In addition, a study by Burstein reveals that treatment with varenicline in the elderly did not increase side effects compared to placebo [20], encouraging the varenicline use in this age group.

As shown in Table 2, there were no differences among the baseline characteristics between quitter and non-quitter groups except for the type of medication used. Smokers receiving varenicline were more likely to achieve cessation than those using NRT. This finding emphasizes the effectiveness of different drugs for smoking cessation in the elderly. Unlike a previous study that shows gender as a predictor of smoking cessation in the elderly [21], we found no associations between gender and smoking cessation success. In addition, other studies show that fewer smoking years and higher daily cigarette amounts are associated with higher smoking cessation rates in the elderly [7]. In our study, we defined “smoking pack-years” as the multiplication of the “daily smoking amount” and “smoking years”, but we found that “smoking pack-years” were not significantly different between quitters and non-quitters. These findings indicated that despite high nicotine dependence score or longer smoking history in older smokers, they still have high potential for quitting smoking. Our study also found that pharmacotherapy plus one-time individual counseling might not improve the smoking cessation rate compared to pharmacotherapy alone. This result reached similar conclusion to another study [22]. The possible inference was that single counseling may not be enough for significant improvement of the smoking cessation rate in the elderly. Further study of the relationship between pharmacotherapy with intensive professional counseling and smoking cessation success rate in the elderly should be investigated in the future.

In Table 3, after adjusting for factors (age, gender, treatment method, hypertension, daily cigarette count, smoking year, FTND score ≥ 7, and carbon monoxide scores ≥ 18 ppm), the patients who received varenicline had a significantly higher cessation rate than those who received NRT (OR, 3.22; 95%CI, 1.23–8.43). The reason why varenicline is more effective than NRT in the elderly is not yet entirely understood. Previous studies have shown that there is no age difference in the pharmacodynamics of varenicline. We proposed that the possible mechanism may be due to the decrease in transdermal nicotine patch absorption in the elderly due to reduced blood perfusion in skin tissue [23]. In addition, previous studies including the EAGLES trial and a meta-analysis, reported that varenicline was more effective than NRT in the general population [12,24]. In our study, we also observed similar results in the elderly population. After reviewing previous studies, there is only one study published in Taiwan comparing the effects of varenicline and NRT in the elderly [25], which had contrary conclusion to our study: varenicline was less effective than NRT in smokers >55 years of age at a six-month follow up (OR 0.75, 95% CI: 0.60–0.94). However, in the study, varenicline was prone to be prescribed to those with higher nicotine dependence scores, which would bias the abstinence rate of varenicline to a lower level. While in our study, there were similar baseline conditions in two groups of medicinal users, which can better compare the effects of these drugs. Therefore, further studies on this topic are warranted in the future to validate these results.

Our study highlighted the importance of the medication treatment for smoking cessation in elderly patients. One of the main advantages of our study was that the decision of medication type was made together by physicians and patients. The patients’ baseline conditions, including the severity of nicotine dependence, did not affect the choice of medications. Therefore, our finding can be applied to the real world. Second, we used not only FTND score, but also “pack-year” and CO concentration as variables to compare the severity of nicotine dependence in quitters and non-quitters. However, we are aware of some limitations in this study. First, our sample size was relatively small, and further large studies are needed to confirm our results. Second, we compared the effects of the two drugs in our study without using placebo as a control group, which may affect the interpretation of the test results. Third, our research was a retrospective cohort study, not a randomized control trial, there may be residual confounding factors, such as alcohol use, educational status as well as marital status. Fourth, our abstinence rate was collected by three-month and six-month quit rate by telephone, which was not confirmed biochemically. Therefore, whether participants quit smoking may be misreported or mis-recorded, resulting in misclassification. However, the NRT and varenicline groups have similar misclassification possibilities, which may not affect the odds ratio of smoking cessation success rates between the two groups.

## 5. Conclusions

Overall, we found that varenicline seemed better than NRT for smoking cessation in the elderly. Other baseline characteristics, such as gender, smoking history, nicotine dependence level, baseline CO concentration, and past medical history may not affect smoking cessation success rates in the elderly.

## Figures and Tables

**Table 1 ijerph-16-03462-t001:** Participants’ baseline characteristics.

	Total n = 129	Varenicline n = 101 (78.3%)	NRT n = 28 (21.7%)	
Variable	Mean ± SD or N (%)	Mean ± SD or N (%)		*p*-Value *
Demographics				
Age (years)	65.9 ± 5.7	66.0 ± 5.9	65.4 ± 5.1	0.58
Men	109 (84.5)	83 (82.2)	26 (92.9)	0.17
Weight (kg)	68.5 ± 11.1	68.4 ± 11.7	68.9 ± 9.1	0.83
Smoking year (years)	40.9 ± 11.1	40.6 ± 11.1	41.9 ± 11.3	0.59
Cigarettes/days	21.8 ± 11.1	22.0 ± 10.5	20.8 ± 13.3	0.59
Pack-years	43.7 ± 23.5	44.4 ± 23.6	41.3 ± 23.5	0.53
Pack-years ≥ 30	94 (72.9)	73 (72.3)	21 (75.0)	0.77
Exhaled CO (ppm)	14.2 ± 8.5	14.6 ± 8.6	13.0 ± 8.2	0.39
CO ≥ 18 ppm	35 (27.1)	29 (28.7)	6 (21.4)	0.44
FTND score	6.2 ± 2.6	6.0 ± 2.5	6.8 ± 2.6	0.12
FTND Score ≥ 7	59 (45.7)	42 (41.6)	17 (60.7)	0.07
Treatment method				
Pharmacotherapy	56 (43.4)	40 (39.6)	16 (57.1)	0.10
Pharmacotherapy with individual counseling	73 (56.6)	61 (60.0)	12 (42.9)	
Comorbidities				
Hypertension	65 (50.4)	51 (50.5)	14 (50.0)	0.96
Diabetes	30 (23.3)	25 (24.8)	5 (17.9)	0.44
Hyperlipidemia	50 (45.0)	46 (45.5)	12 (42.9)	0.80
Heart disease	31 (24.0)	27 (26.7)	4 (14.3)	0.17
Lung disease	15 (11.6)	9 (8.9)	6 (21.4)	0.07
Neurological disease	17 (13.2)	13 (12.9)	4 (14.2)	0.84
Cancer	6 (4.7)	4 (4.0)	2 (7.1)	0.48
Psychiatric disease	16 (12.4)	11 (10.9)	5 (17.9)	0.32

Abbreviations: CO = carbon monoxide; FTND score = Fagerström test for nicotine dependence score; NRT = nicotine replacement therapy; SD = standard deviation; * using chi-squared and *t*-test.

**Table 2 ijerph-16-03462-t002:** Comparison of baseline characteristics related to smoking cessation among quitters and non-quitters.

	Quitters (n = 62)	Non-Quitters (n = 67)	*p*-Value *
Variable	Mean ± SD or N (%)		
Demographics			
Age (years)	66.3 ± 6.8	65.5 ± 4.4	0.47
Men	53 (85.5)	56 (83.6)	0.77
Weight (kg)	68.1 ± 11.7	68.9 ± 10.7	0.67
Smoking year (years)	40.7 ± 11.7	41.0 ± 10.7	0.86
Cigarettes per day	21.0 ± 10.0	22.5 ± 12.1	0.46
Pack-years	41.4 ± 20.8	45.8 ± 25.7	0.29
Pack-year ≥ 30	47 (75.8)	47 (70.2)	0.47
Exhaled CO (ppm)	13.6 ± 8.4	14.8 ± 8.7	0.42
CO ≥ 18 ppm	16 (25.8)	19 (28.4)	0.74
FTND score	6.1 ± 2.5	6.2 ± 2.6	0.75
FTND Score ≥7	27 (43.6)	32 (47.8)	0.63
Drug type			
Varenicline	54 (87.1)	47 (70.2)	0.02
NRT	8 (12.9)	20 (29.9)	
Treatment method			
Pharmacotherapy	25 (40.3)	31 (46.3)	0.50
Pharmacotherapy with individual counseling	37 (59.7)	36 (53.7)	
Drug duration (week)	5.5 ± 4.4	6.5 ± 4.9	0.21
Comorbidities			
Hypertension	34 (54.8)	31 (46.3)	0.33
Diabetes	11 (17.7)	19 (28.4)	0.15
Hyperlipidemia	26 (41.9)	32 (47.8)	0.51
Heart disease	14 (22.5)	17 (54.8)	0.71
Lung disease	7 (11.3)	8 (11.9)	0.91
Neurological disease	7 (11.3)	10 (14.9)	0.54
Cancer	4 (6.5)	2 (3.0)	0.35
Psychiatric disease	5 (8.1)	11 (16.4)	0.15

Abbreviations: CO = carbon monoxide; FTND = Fagerström test for nicotine dependence; NRT = nicotine replace therapy; SD = standard deviation; * using chi-squared and *t*-test.

**Table 3 ijerph-16-03462-t003:** Odds ratios (95% confidence interval) of successful smoking abstinence by logistic regression models.

	Model 1	Model 2
	OR (95% CI)	OR (95% CI)
Age, years	1.02 (0.96–1.09)	1.01 (0.94–1.08)
Gender (men/women)	1.16 (0.44–3.01)	1.24 (0.45–3.42)
Treatment method (Pharmacotherapy + Individual counseling/Pharmacotherapy)	1.27 (0.63–2.56)	1.19 (0.55–2.60)
With hypertension	1.41 (0.71–2.82)	1.49 (0.71–3.12)
Daily cigarette amount	0.99 (0.96–1.02)	0.98 (0.94–1.02)
Smoking year	0.99 (0.97–1.03)	0.99 (0.96–1.03)
FTND ≥7	0.84 (0.42–1.69)	1.07 (0.90–1.27)
Carbon monoxide ≥18	0.88 (0.40–1.91)	0.98 (0.94–1.03)
Drug type (Varenicline/NRT)	2.87 (1.16–7.12)	3.22 (1.23–8.43)

Model 1: univariate logistic regression. Model 2: multivariate logistic regression. Abbreviations: FTND = Fagerström test for nicotine dependence; NRT = nicotine replace therapy.

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
