# Peer review of "Factors Correlated with Smoking Cessation Success in Older Adults: A Retrospective Cohort Study in Taiwan"

_ijerph, 2019, doi:10.3390/ijerph16183462_

Round 1

Reviewer 1 Report

The article concerns a very important issue, however the title is too general and "redundant" (it promises what is not in the article). The authors wrote: "We designed a study on smoking cessation predictors in elderly patients, and compared different treatment types in an outpatient care unit to provide effective information for improving cessation rates among the elderly". but they did not indicate any predictors of quitting smoking. In the restrictions they rightly indicated that they did not have a control group (they did not use a placebo), they did not ask about alcohol consumption, education, marital status. Their analysis does not show which elderly people need special attention by suggesting them to quit smoking

In fact, the article is only about comparing the effectiveness of two drugs to help stop smoking in people over 60 years of age. and that's why that's all that should be in the title.

In addition, the methodology implemented is not very sophisticated.

It should be clarified whether it is known why some patients received NRT and some varenikline if group selection was intentional, what factors determined it, whether the underlying causes were, for example, the occurrence of certain chronic diseases, mental diseases and if so, it could affect the result of the examination.

The section on study limitations omits information that tobacco abstinence data has not been objectively verified and how it affects the study results.

Both the discussion and the section devoted to the limitations of the study require extension and careful discussion.

Given the methodological limitations, I suggest considering publishing art. as short communication instead of full length original article.

Reviewer 2 Report

1. I think that the introduction of the novelty of this research is insufficient in the introduction. Can you add where the novelty is?
2. L98-101 Is not it necessary to adjust the smoking years (period)?
3. Is L103 “Our criterion for statistical significance was p <0.05.” Required? Isn't it overlapping with the text after that?
4. It is not appropriate to be in the section of L105-106 Statistical analysis. How about making another section as 2.4.  Ethical approval?
5. All tables should be created so that they can be used independent. It is necessary to describe in all tables what the numerical value (mean ± SD) and abbreviations represent.
6. For Tables 1 and 2, you can consider submitting this as Supplement files. This gives a redundant impression. However, if you think you need it, you can leave it in the text as it is.
7. Please add “baseline” to the title in Table 2. This is baseline characteristics, right?
8. Delete some of the traces of the calibration.

Round 2

Reviewer 1 Report

The Authors have improved the manuscript but it still need to be amended. They added indications / contraindications for the use of Varenicline or NRT. But I noticed something that worried me. Well, in line 74 it says "The treatment methods included drug with or without educational interventions", while further on the effects of educational intervention or its absence I found nothing. To be precise, the word "educational" is repeated only in the tables, but not once in the text. And something could be said reasons why education is not working (?). At the same time, the Authors write that their older people were highly motivated to quit smoking, where did this motivation come from? So alone, without any suggestion, they were eager to quit smoking and were only looking for support in medication. In addition, why separate Table 1 with the characteristics of the group, and then Table 3 divided into two groups. You can make one table and three columns, which is a waste of space.

Round 3

Reviewer 1 Report

i have no further comments